# Controlling chaos using edge computing hardware

Robert M. Kent [1], Wendson A. S. Barbosa [1] & Daniel J. Gauthier [1,2] ✉

Machine learning provides a data-driven approach for creating a digital twin of a system – a digital model used to predict the system behavior. Having an accurate digital twin can drive many applications, such as controlling autonomous systems. Often, the size, weight, and power consumption of the digital twin or related controller must be minimized, ideally realized on embedded computing hardware that can operate without a cloud-computing connection. Here, we show that a nonlinear controller based on next-generation reservoir computing can tackle a difficult control problem: controlling a chaotic system to an arbitrary time-dependent state. The model is accurate, yet it is small enough to be evaluated on a field-programmable gate array typically found in embedded devices. Furthermore, the model only requires $25.0 \pm 7.0$ nJ per evaluation, well below other algorithms, even without systematic power optimization. Our work represents the first step in deploying efficient machine learning algorithms to the computing "edge."

We are now in the age of artificial intelligence and machine learning (ML), which is leading to disruptive technologies such as natural language processing[1], system design optimization[2], and autonomous vehicles[3]. With these impressive advances, there is a growing interest in applying modern ML tools to existing problems to improve performance and give new insights. One area of recent success is developing advanced controllers for complex dynamical systems, which involves controlling a device or collection of interacting systems that vary in time, such as advanced aircraft, robots, electric motors, among others.

A controller automatically adjusts accessible system parameters based on real-time measurement of the system to guide it to a desired state. Traditional and widely deployed control methods assume that the system responds in a linear fashion to a control adjustment, referred to as a linear controller. However, most systems have a nonlinear response, thereby requiring a precise mathematical model of the system or thorough system characterization and adjusting the linear controller on the fly based on this information.

A particular difficult control task is to stabilize a nonlinear system displaying a complex behavior known as chaos, which is characterized by sensitivity to initial conditions and tiny perturbations. Previous approaches, such as that pioneered by Ott, Grebogi, and Yorke (OGY)[4]

and by Pyragas[5], are restricted to stabilizing unstable states embedded in the dynamics and are restricted to making small adjustments to the accessible system parameters and hence control can only be turned on once the system is close to a desired state.

Machine-learning-based controllers offer a new approach that has the advantage of learning the nonlinear properties of the system based only on data, relearning the system dynamics to respond to system degradation over time[6] or damage events[7], and monitoring system health and status[8]. However, recent ML solutions are often computationally expensive, requiring computational engines such as power-hungry graphical processor units, or developing custom neuromorphic chips using analog-digital designs or advanced materials that operate at low power[9–11].

An alternative approach is to simplify the ML algorithm so that it can be deployed on commercial off-the-shelf microcontrollers or field-programmable gate arrays (FPGAs). These compute devices are small and consume low power, thus making them well-suited for edge-computing and portable devices without requiring a connection to cloud-computing resources. One candidate approach is to use a reservoir computer[12,13] (RC), a best-in-class ML algorithm for learning dynamical systems. Previous work by Canaday et al.[14] used a RC to learn an inverse-based control algorithm, but it failed to achieve

[1]The Ohio State University, Department of Physics, 191 West Woodruff Ave., Columbus, OH 43210, USA. [2]ResCon Technologies, LLC, 1275 Kinnear Rd., Suite 239, Columbus, OH 43212, USA. ✉e-mail: gauthier.51@osu.edu

accurate control. We believe these shortcomings are due to the learned inverse not being unique or due to control-loop latency that was not learned.

To avoid these problems, we use a feedback linearization algorithm combined with a highly efficient next-generation reservoir computer (NG-RC)[15] that greatly simplifies the controller and reduces the computational resources. Here the NG-RC predicts the future state of the system, which is feedback via the controller to cancel the nonlinear terms of the system's nonlinear dynamical evolution while simultaneously providing linear feedback to stabilize the system to the desired time-dependent state.

We demonstrate that an NG-RC-based controller can stabilize the dynamics of a chaotic electronic circuit[16] to trajectories that are impossible for classic chaos control methods and has lower error than the inverse-control algorithm used by Canaday et al.[14] as well as a linear controller. We implement our design using a FPGA on a commercial demonstration board to collect real-time data from the circuit, process the data using a firmware-based NG-RC, and apply the control perturbations. An FPGA has the advantage that it can perform parallel computations well matched to our ML algorithm and it has memory co-located with the processing logic gates, which is known to boost efficiency[10,11]. Furthermore, the ML algorithm is so simple and efficient that its execution time is much faster than other system processes and its power consumption is a tiny fraction of the overall power budget.

Below, we give an overview of our experimental system, introduce the control algorithm, and provide an overview of reservoir computing. We then present our results on suppressing chaos in the circuit by guiding the system to desired states and give key performance metrics. We close with a comparison to other recent ML-based control approaches and discuss how our controller can be further improved with optimized hardware.

## Results

An overview of our prototype chaotic circuit and nonlinear control system is shown in Fig. 1. The chaotic circuit (known as the "plant" in the control literature) consists of passive components including diodes, capacitors, resistors, and inductors, and an active negative resistor realized with an operational amplifier that can source and sink power to the rest of the circuit. The phase space of the system is three-dimensional specified by voltages $V_1$ and $V_2$ across two capacitors in

the circuit and the current $I$ passing through the inductor (see Methods). To display chaos, the circuit must be nonlinear, which arises from the diodes as evident from their nonlinear current-voltage relation. We adjust the circuit parameters so that it displays autonomous double-scroll chaotic dynamics characterized by a positive Lyapunov exponent, indicating chaotic dynamics[16]. A two-dimensional projection of the phase space trajectory allows us to visualize the corresponding strange attractor and is show in Fig. 1.

To control the system, we measure in real time two accessible variables – voltages $V_1$ and $V_2$ – using analog-to-digital converters located on the FPGA (the sensors indicated in Fig. 1). We are limited to measuring two variables because of the available resources on the device. For a similar reason, we are limited to controlling a single variable, which we take as $V_1$ based on previous control experiments with this circuit[16]. Here, the user provides a desired value $V_{1,des}$ as the control target, which can be time-dependent.

The goal of the controller is to guide the system to the desired state by perturbing its dynamics based on a nonlinear control law. Controlling this system to a wide range of behaviors requires that the controller also be nonlinear; the applied perturbations are a nonlinear function of the measured state variables. Importantly, a nonlinear controller allows us to go beyond established chaos control methods that are restricted to controlling about sets embedded in the dynamics and only applying control when the system is close to one of these sets[4,17,18], as mentioned above.

Coming back to the control implementation, we use two phases: (1) a learning phase where we learn the dynamics of the chaotic circuit and how it responds to random perturbations (Fig. 1, top panel) in an "open" configuration; and (2) a closed-loop configuration (Fig. 1, bottom panel), where the learned model generates perturbations based on the real-time measurements and the user-specified desired state. We store the desired state in local on-chip memory to reduce the system power consumption, but it is straightforward to interface our controller with a higher-level system manager or user interface to allow greater flexibility.

During the training phase, we synthesize a low-pass-filtered random sequence and store it in on-chip memory (see Methods). This sequence is read out of memory and sent off-chip to a digital-to-analog converter, which passes to a custom-built voltage-to-current converter ($u_1$ in the top panel of Fig. 1, red waveform) and injected into the

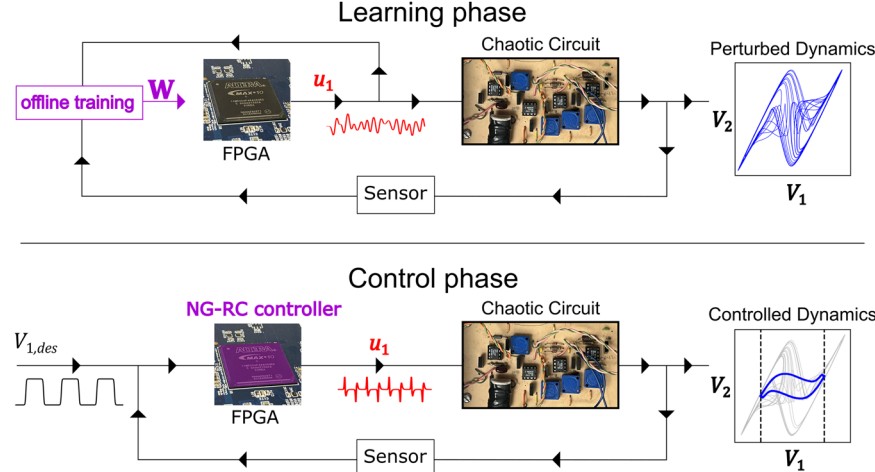

**Fig. 1 | NG-RC based nonlinear controller realized on edge computing hardware.** (top) Learning phase. A field programmable gate array (FPGA) applies perturbations (red) to a chaotic circuit and the perturbed dynamics of the circuit (blue) are measured by a sensor (analog-to-digital converters). The temporal evolution of the perturbations and responses are transferred to a personal computer (left, purple) to learn the parameters of the NG-RC controller **W**. These parameters are programmed onto the FPGA as well as the firmware for the controller. (bottom) Control phase. The NG-RC controller implemented on the FPGA measures the dynamics of the chaotic circuit with a sensor (analog-to-digital converters) in real time and receives a desired trajectory $V_{1,des}$ for the $V_1$ variable, and computes a suitable control signal (red) that drives the circuit to the desired trajectory.

chaotic circuit at the node corresponding to the capacitor giving rise to voltage $V_1$.

The control current perturbs the chaotic dynamics so we can learn how it responds to perturbations at many locations in phase space, which is common in extremum-seeking control[19], for example. To this end, we simultaneously measure the voltages $V_1$ and $V_2$ and store their values in on-chip memory. After a training session, the data is transferred to a laptop computer to determine the NG-RC model as described below. The model is then programmed in the FPGA logic.

Nonlinear control is turned on by "closing the loop." Here, the real-time measured voltages and the desired state are processed by the NG-RC model, control perturbations calculated, and subsequently injected into the circuit. There is always some latency between measurements and perturbation, which can destabilize the control if it becomes comparable to or larger than the Lyaponov time – the interval over which chaos causes signal decorrelation[20]. We optimize the overall system timing to minimize the latency, but the NG-RC model and control law is so simple that the evaluation time is two orders of magnitude shorter than the other latencies present in the controller, which are detailed Supplementary Note 7. Another important aspect of our controller is that we use fixed-point arithmetic, which is matched to the digitized input data and reduces the required compute resources and power consumption[10].

## Control law
We formalize the control problem as follows: the accessible variables of the chaotic circuit $\mathbf{X} \in \mathbb{R}^{d'}$ are the inputs to a nonlinear control law that specifies the control perturbations $\mathbf{u} \in \mathbb{R}^d (d \le d')$ necessary to guide a subset of the accessible system variables $\mathbf{Y} \in \mathbb{R}^d$ to a desired state $\mathbf{Y}_{des} \in \mathbb{R}^d$. We assume the dynamics of the system evolve in continuous time, but discretely sampled data allows us to describe the dynamics of the uncontrolled system in the absence of noise by the mapping

$$\mathbf{X}_{i+m} = \mathbf{F}'_{i+m}\left(\mathbf{X}_i, \mathbf{X}_{i-1}, \dots, \mathbf{X}_{i-(k-1)}\right), \tag{1}$$

where $i$ represents the index of the variables at time $t_i$, $m$ is the number of timesteps the map projects into the future and $\mathbf{F}'_{i+m} \in \mathbb{R}^{d'}$ is a nonlinear function specifying the flow of the system, which may depend on $k$ past values of the system variables. The dynamics of the controlled variables are then given by $\mathbf{Y}_{i+m} = \mathbf{F}_{i+m}$, where $\mathbf{F}_{i+m} \in \mathbb{R}^d$ is related to $\mathbf{F}'_{i+m}$ through a projection operator. We assume the control perturbations at time $t_i$ affect the dynamics linearly, while past values may have nonlinear effects on the flow of the system, allowing us to express the dynamics of the controlled system variables as

$$\mathbf{Y}_{i+m} = \mathbf{F}_{i+m}\left(\mathbf{X}_i, \mathbf{X}_{i-1}, \dots, \mathbf{X}_{i-(k-1)}, \mathbf{u}_{i-1}, \dots, \mathbf{u}_{i-(k-1)}\right) + \mathbf{W}_u \mathbf{u}_i, \tag{2}$$

where we omit the arguments of $\mathbf{F}_{i+m}$ below for brevity.

To control a dynamical system of the form (2), we use a general nonlinear control algorithm developed for discrete-time systems[21]. Different from many previous nonlinear controllers[22], we do not require a physics-based model of the plant. Rather, we use a data-driven model learned during the training phase. Robust and stable control of the dynamical system is obtained[21,23] by taking

$$\mathbf{u}_i = \hat{\mathbf{W}}_u^{-1}\left[\mathbf{Y}_{des,i+m} - \hat{\mathbf{F}}_{i+m} + \mathbf{K}\mathbf{e}_i\right], \tag{3}$$

where, $\mathbf{Y}_{des,i+m} \in \mathbb{R}^d$ is the desired state of the system $m$-steps-ahead in the future, $\mathbf{K} \in \mathbb{R}^{d \times d}$ is a closed loop gain matrix, and $\mathbf{e}_i = \mathbf{Y}_i - \mathbf{Y}_{des,i}$ is the tracking error. The "^" over the symbols indicate that these quantities are learned during the training phase using the procedure described in the next subsection. A key assumption is that we can learn $\mathbf{F}_{i+m}$ only using information from the accessible system variables and past perturbations.

The control law (3) executes feedback linearization, where the feedback attempts to cancel the nonlinear function $\mathbf{F}_{i+m}$ in mapping (2), thus reducing to a linear control problem. To see this, assume perfect learning ($\hat{\mathbf{F}}_{i+m} = \mathbf{F}_{i+m}$, $\hat{\mathbf{W}}_u = \mathbf{W}_u$), and apply the definition of the tracking error to obtain

$$\mathbf{e}_{i+1} = \mathbf{K}\mathbf{e}_i. \tag{4}$$

This system is globally stable when all eigenvalues of $\mathbf{K}$ are within the unit circle. Adjustments to $\mathbf{K}$ are required to maintain globally stable control in the presence of bounded modeling error and noise[21].

For comparison, we compare our nonlinear controller to a simple linear proportional feedback controller. We can make a simple adjustment to the control law described above to realize a linear controller. Consider the case when the sampling time t is short relative to the timescale of the system's dynamics, allowing us to approximate the flow $\hat{\mathbf{F}}_{i+m}$ in the control law (3) with the current state $\mathbf{Y}_i$ to arrive at a linear control law. This is motivated by approximating the flow using Euler integration with a single step $\mathbf{Y}_{i+1} \approx \mathbf{Y}_i + t\mathbf{f}(t_i, \mathbf{X}_i)$, where $\mathbf{f}(t_i, \mathbf{X}_i)$ is the vector field governing the dynamics, and observing that $\mathbf{Y}_{i+1} \approx \mathbf{Y}_i$ when t approaches zero.

Based on the number of accessible system variables and control perturbations in our system, we take $d' = 2$ and $d = 1$ corresponding to $\mathbf{X}_i = [V_{1,i}, V_{2,i}]^T$, $\mathbf{Y}_i = V_{1,i}$, $\mathbf{u}_i = u_{1,i}$, and the feedback control gain $K$ is a scalar. Additionally, we take $m = 1$ (2) in this work for one-step- (two-step-) ahead prediction, as this provides a trade-off between controller stability and accuracy.

## Reservoir computing
Here, we briefly summarize the NG-RC algorithm[12,13] used to estimate $\hat{\mathbf{W}}_u^{-1}$ and $\hat{\mathbf{F}}$ needed for the control law. Like most ML algorithms, an RC maps input signals into a high-dimensional space to increase computational capacity, and then projects the system onto the desired low-dimensional output. Different from other popular ML algorithms, RCs have natural fading memory, making them particularly effective at learning the behavior of dynamical systems.

A RC gains much of its strength by using a linear-in-the-unknown-parameters universal representation[24] for $\hat{\mathbf{F}}$, which vastly simplifies the learning process that uses linear regularized least-squares optimization. The other strength is that the number of trainable parameters in the model is smaller than other popular ML algorithms, thus reducing the size of the training dataset and the compute time for training and deployment. Previous studies have successfully used an RC to control a robotic arm[25] and a chaotic circuit[14], for example, using an inverse-control method[26].

The next-generation reservoir computing algorithm has similar characteristics to a traditional RC[27], but has fewer parameters to optimize, requires less training data, and is less computationally expensive[15]. In this framework, the function $\hat{\mathbf{F}}$ is represented by a linear superposition of nonlinear functions of the accessible system variables $\mathbf{X}$ at the current and past times, as well as perturbations $\mathbf{u}$ at past times, as mentioned previously. The number of time-delayed copies $k$ of these variables form the memory of the NG-RC. Based on previous work on learning the double-scroll attractor[15], we use $k = 2$ and use odd-order polynomial functions of $\mathbf{X}$ up to cubic order, but we only take linear functions of past perturbations $\mathbf{u}_{i-1}$ for simplicity. The feature vector also contains the perturbations $\mathbf{u}_i$, which allows us to estimate $\hat{\mathbf{W}}_u$. We perform system identification[28,29] to select the most important components of the model. Additional details regarding the NG-RC implementation are given in the Methods.

## Control tasks
We apply our nonlinear control algorithm with three challenging tasks. The first is to control the system to the unstable steady state (USS) located at the origin $[V_1 = 0, V_2 = 0, I = 0]$ of phase space starting from a

random point located on the strange attractor. Previous work by Chang et al.[16] fail to stabilize the system at the origin using an extended version of the Pyragas approach[30].

The second task is to control the system about one of the USSs in the middle of one of the "scrolls" and then guide the system to the USS at the other scroll and then back in rapid succession. Rapid switching between these USSs is unattainable using Pyragas-like approaches because the controller must wait for the system to cross into the opposite basin of attraction defined by the line $V_1 = 0$[16], which cannot occur within the requested transition time. Additionally, the system never visits the vicinity of these USSs[16] and hence large perturbations are required to perform the fast transitions, causing the OGY[4] approach to fail.

Last, we control the system to a random trajectory, which determines if the system can be controlled to an arbitrary time-dependent state. This task is difficult because the controller must cancel the nonlinear dynamics at all points in phase space, requiring an accurate model of the flow. This task is infeasible for Pyragas-like approaches, as they are limited to controlling the system to UPOs and USSs. There are extensions of the OGY method that can steer the system to arbitrary points on the attractor[31], but controlling between arbitrary points quickly requires large perturbations, making this approach unsuitable. Furthermore, this approach can require significant memory to store paths to the desired state[32], increasing the power and resource consumption.

### Example trajectories

Example phase-space trajectories of the controlled system for the three different tasks are shown in Fig. 2. For the first, the dynamics of

the system are far from the origin when the controller is turned on (indicated by an × in the phase portrait) and quickly evolves toward the origin and remains stable for the remainder of the control phase. Note that we only control $V_1$; $V_2$ and $I$ (not shown) are naturally guided to zero through their coupling to $V_1$.

For the second task, it is seen that $V_1$ and $V_2$ are successfully guided to the corresponding values of the USSs. The trajectory followed by the system as it transitions between the USSs depends on the requested transition rate because changes in $V_2$ lags changes in $V_1$ as governed by their nonlinear coupling. The temporal evolution of the system is shown in Fig. 2d, e, where we see that $V_1$ accurately follows the desired trajectory without any visible overshoot.

For the final task, we request that $V_1$ follows a random trajectory contained within the domain of the strange attractor as shown in Fig. 2c; additional results visualizing the controlled system is given in Supplementary Note 2.

### Control performance

We now characterize the stability of the nonlinear controller as a function of the feedback gain $K$ and compare the control error on the three tasks for the linear and nonlinear controllers. Figure 3 shows the performance of the controller as $K$ varies, where Fig. 3a shows the error between the requested and actual state of the system, while Fig. 3b shows the size of the control perturbations. We see that there is a broad range of feedback gains that give rise to high-quality control, and the control perturbations are small at the USSs, showing the robustness of our approach. The domain of control is somewhat smaller than that expected, which we believe is due to small modeling

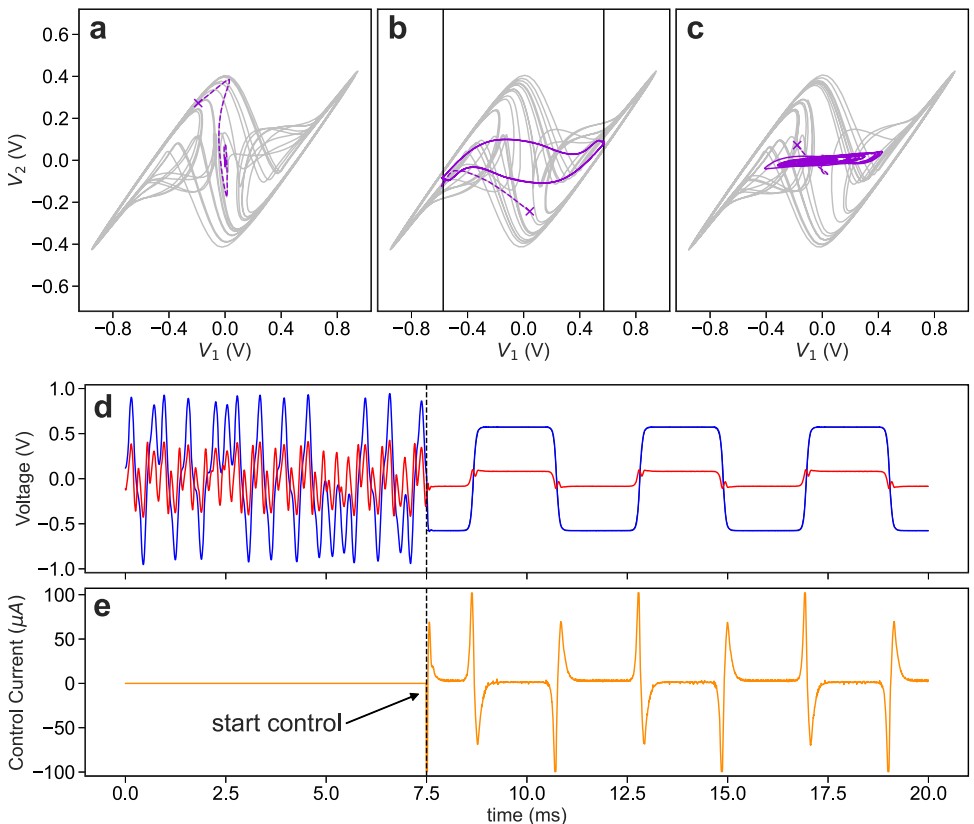

**Fig. 2 | Controlling a dynamical system using the NG-RC for one-step-ahead prediction.** Controlled attractors for **a** the first task controlling the system to the origin, **b** controlling back and forth between the two USSs (solid black), and **c** controlling to a random waveform. The unperturbed attractor (gray) before the control is switched on, the moment the control is switched on (purple ×), the transient for the system to reach the desired trajectory (dashed purple), and the

controlled system (solid purple). **d** Temporal evolution of $V_1$ (blue), and $V_2$ (red) during the second task. When the control begins, the chaotic system follows the desired trajectory (solid black), which is under the blue curve. **e** The control perturbation before and after the control is switched on (orange). The control gain is set to $K = 0.75$ for all cases.

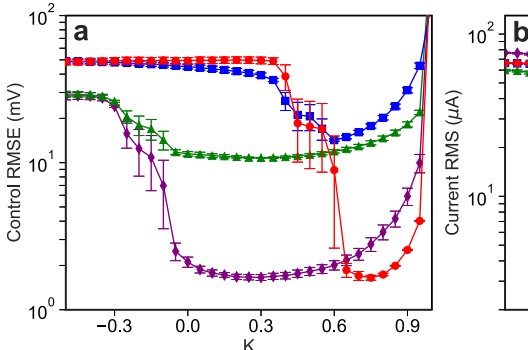
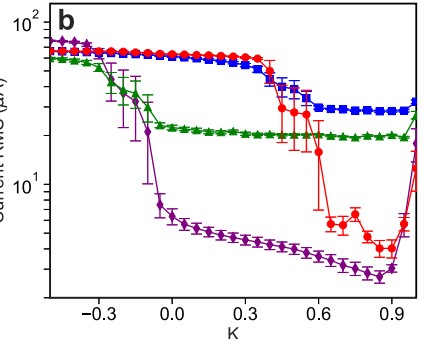

**Fig. 3 | Performance of the one-step ahead NG-RC controller. a** RMSE of the control and **b** RMS of the control current when controlling the system to the phase-space origin (red circles), back-and-forth between two USSs (green triangles), to a random waveform (blue squares), and stabilized at either nonzero USS (purple diamonds) as a function of the control gain $K$.

**Table 1 | Minimum control error**

| Controller type | Origin task | Two USS task | Two USS task (no transients) | Random waveform task |
|---|---|---|---|---|
| Linear | $2.61 \pm 0.11$ mV | $11.98 \pm 0.51$ mV | $1.67 \pm 0.05$ mV | $25.10 \pm 1.90$ mV |
| one-step ahead | $1.65 \pm 0.06$ mV | $10.79 \pm 0.07$ mV | $1.65 \pm 0.08$ mV | $14.23 \pm 0.24$ mV |
| two-step ahead | $1.64 \pm 0.02$ mV | $11.23 \pm 0.26$ mV | $1.95 \pm 0.11$ mV | $14.57 \pm 0.44$ mV |

The minimum RMSE for each controller type and control task with uncertainty derived from the standard error over five trials.

errors and latency between measuring the state and applying the control perturbation[20,33]. We find that the NG-RC controller with two-step-ahead prediction has a substantially larger domain of control, which is detailed in Supplementary Note 3.

The minimum error for each task and controller type is given in Table 1. We see that the NG-RC controllers outperforms the linear controller on every task, except when the system is stabilized at the nonzero USSs because the dynamics are approximately linear there.

The advantage of the NG-RC controller is evident for the random waveform task, where we observe a $1.7\times$ reduction in the error compared to the linear controller. This task is especially challenging for the linear controller because the desired state visits a wide range of phase space with different local linear dynamics and the sampling rate is too slow for $\mathbf{Y}_{i+1} \approx \mathbf{Y}_i$ to be a good approximation, making the controller ineffective at canceling the nonlinear dynamics of the system. Additional performance metrics, such as the error, size of control perturbations, and sensitivity to control-loop latency are given in the Supplementary Note 3.

### Control resources

We now discuss the resources needed for implementing control. First, the FPGA demonstration board has resources well matched to the chaotic system. The characteristic timescale of the chaotic circuit is 24 μs, which compares favorably with the 1 Msample/s sampling rate of the analog-to-digital and digital-to-analog converters and our control update period of 5 μs. Unlike previous ML-based controllers, the update period is not limited by the evaluation time of the ML model and control law, which only requires a computation time of 50 ns.

Regarding power consumption, our demonstration board is not designed for low-power operation and there is a substantial power-on penalty when using the board. For the NG-RC-based controllers, we observe a total power consumption of $1384.0 \pm 0.7$ mW, where $53.5 \pm 1.4$ mW is consumed by the analog-to-digital and digital-to-analog converters, $5.0 \pm 1.4$ mW is consumed evaluating the NG-RC algorithm, and the remainder is for other on-board devices not used directly by the control algorithm. Considering the control loop rate of 200 kHz, these results

indicate that we only expend $25.0 \pm 7.0$ nJ per inference if we consider only the energy consumed by the NG-RC, or $6920.0 \pm 3.5$ nJ considering the total power. In contrast, the linear controller requires only $7.5 \pm 7.0$ nJ per inference, which is primarily due to the decreased number of multiplications.

Regarding the logic and math resources on the FPGA, the one-step ahead controller uses 1722 logic elements (3% of the available resources), 924 registers (1.8%), and 18 multipliers (13%). The two-step ahead controller requires 16 additional registers to store the desired state two steps in the future, but otherwise uses the same resources. In contrast, the linear controller uses 986 (2%) logic elements, 604 registers (1.2%), and 4 multipliers. All controllers use only 768 memory bits (< 1%) for the origin task, which does not require any additional memory to store the desired signal. These resources are small in comparison to other recent ML-based control algorithms evaluated on an FPGA, which we present in Supplementary Note 4. Additionally, we show that the computational complexity of our approach is significantly lower than the RC-based inverse controller by Canaday et al.[14] in Supplementary Note 1.

## Discussion

We provide a proof-of-concept demonstration of using a state-of-the-art ML algorithm to control a complex dynamical system deployed on a small size and low power embedded computing device. The nonlinear controller uses very few resources when implemented on an FPGA, it's highly parallelizable resulting in an inference time on the nanosecond timescale, and it is robust to modeling and discretization error, delays, and noise. Even though we do not use a power-efficient device, it consumes less energy per inference than most ML-based controllers.

When compared to a linear controller, our approach achieves a $1.7\times$ smaller error on the most difficult control task, requiring only a small increase in the percentage of available FPGA resources and the power consumption. Furthermore, we achieve better performance than the RC-based inverse control law approach by Canaday et al.[14] while using $4\times$ less training data, $6\times$ fewer trainable parameters, $30\times$ fewer multiplications, and 60 fewer evaluations of the hyperbolic tangent function. Additionally, we use 18-bit arithmetic rather than 32-

bit arithmetic, which uses fewer resources on the FPGA device. These differences demonstrate that a nonlinear controller based on the NG-RC algorithm achieves higher performance using substantially few compute resources.

There are several directions to improve upon our results. One is to perform the training directly on the FPGA using on-line regularized regression, as mentioned above. Another is to apply the controller to higher-dimensional problems such as spatial-temporal dynamics[34]. Regarding the power consumption, it can be likely be reduced to the sub-mW level using a more power-efficient FPGA on a custom circuit board without extraneous components. Because our algorithm only requires a small number of multiplications and additions, the model can be evaluated efficiently on the recent custom artificial-intelligence processors[10,11] or using a custom application-specific integrated circuit to achieve the lowest power consumption.

## Methods
### Next generation reservoir computing
We use a next-generation reservoir computer to estimate $\mathbf{F}$ and $\mathbf{W}_u$. The first step is to construct a *linear feature vector* containing the measured variables $\mathbf{X}_i$ as well as $k$ time-delays as

$$\mathbb{O}_{lin,i} = \mathbf{X}_i \oplus \mathbf{X}_{i-1} \oplus \mathbf{X}_{i-2} \oplus \dots \oplus \mathbf{X}_{i-(k-1)}. \quad (5)$$

where $\oplus$ is the concatenation operation. To suitably approximate the nonlinearities in $\hat{\mathbf{F}}$, nonlinear functions of $\mathbb{O}_{lin,i}$ are required, which are often taken to be polynomials as they can be used to approximate the dynamics of many systems. After the nonlinear feature vector $\mathbb{O}_{nonlin,i}$ is constructed, all features that relate to $\mathbf{X}_i$ can be combined in the feature vector

$$\mathbb{O}_{\mathbf{X},i} = c \oplus \mathbb{O}_{lin,i} \oplus \mathbb{O}_{nonlin,i}, \quad (6)$$

where $c$ is a constant, which is not required in this case because there is no constant in the differential equations that describe the dynamics of $V_1$ in the chaotic circuit in Eq. (12). The feature vector used to estimate $\mathbf{F}$ is then given by

$$\mathbb{O}_{\mathbf{F},i} = \mathbf{u}_{i-1} \oplus \mathbb{O}_{\mathbf{X},i}, \quad (7)$$

where only a single past perturbation $\mathbf{u}_{i-1}$ is used because we choose $k = 2$. Lastly, we construct the total feature vector

$$\mathbb{O}_{total,i} = \mathbf{u}_i \oplus \mathbb{O}_{\mathbf{F},i}, \quad (8)$$

and learn the weights $\mathbf{W}$ that map $\mathbb{O}_{total,i}$ onto $\mathbf{Y}_{i+m}$, or

$$\mathbf{Y}_{i+m} \approx \mathbf{W}\mathbb{O}_{total,i} = \hat{\mathbf{W}}_{\mathbf{F}}\mathbb{O}_{\mathbf{F},i} + \hat{\mathbf{W}}_u\mathbf{u}_i, \quad (9)$$

where $\hat{\mathbf{W}}_{\mathbf{F}}$ are the weights that relate to the $\mathbf{F}$, and $\hat{\mathbf{W}}_u^{-1}$ can be computed to complete the control law. If learning is perfect, then $\hat{\mathbf{W}}_{\mathbf{F}}\mathbb{O}_{\mathbf{F},i} = \mathbf{F}_{i+m}$ and $\hat{\mathbf{W}}_u = \mathbf{W}_u$. The learning process involves computing $\mathbf{W}$ using ridge regression[35], which has the explicit solution

$$\mathbf{W} = \mathbf{Y}_d\mathbb{O}_{total}^T\left(\mathbb{O}_{total}\mathbb{O}_{total}^T + \alpha\mathbf{I}\right)^{-1}, \quad (10)$$

where $\mathbf{Y}_d$ contains $\mathbf{Y}_{i+m}$, and $\mathbb{O}_{total}$ contains $\mathbb{O}_{total,i}$, for all times $t_i$ in the training data set, and $\alpha$ is the ridge parameter that can be tuned to prevent overfitting.

The learned weights are then used to perform predictions on a validation data set, and the prediction performance is evaluated using the root mean square error

$$\mathrm{RMSE} = \sqrt{\left\langle (x_i - \hat{x}_i)^2 \right\rangle}, \quad (11)$$

where $x_i$ is the true value and $\hat{x}_i$ is the predicted value. The ridge parameter is tuned so that the prediction RMSE on the validation data is minimized. This metric is also used to characterize the control performance, where $x_i$ is the desired signal, and $\hat{x}_i$ is the state of the controlled system.

### Double scroll circuit
The dynamics of the chaotic circuit shown in Fig. 4 are described by the differential equations

$$C_1\dot{V}_1 = V_1/R_n - g(V_1 - V_2) + u_1,$$
$$C_2\dot{V}_2 = g(V_1 - V_2), \quad (12)$$
$$L\dot{I}_1 = V_2 - R_m$$

where $g(V) = V/R_d + 2I_r \sinh(\alpha V/V_d)$, $I_r = 5.63$ nA, $\alpha = 11.6$, $V_d = 0.58$ V, and $u_1$ is the accessible control current. The USSs are [$V_1 = 0.58$ V, $V_2 = 0.08$ V, $I = 0.20$ mA] and [$V_1 = -0.58$ V, $V_2 = -0.08$ V, $I = -0.20$ mA], which were calculated using the Eq. (12).

### Generating training data
To collect the training and validation data required to optimize the NG-RC controller, we perturb the system with a signal that contains 4000 points, and use 2000 points for training and 2000 points for validation. To create the perturbations, we generate uniform random noise with a range of ±1.5 and zero mean using the random.uniform package in NumPy[36]. We then filter the noise using a 10th-order low-pass Butterworth filter with a cut-off frequency of 900 Hz, which is implemented using the signal.butter package in SciPy[37]. The signal is then multiplied by an envelope function so that it can be smoothly repeated on the FPGA if longer training times are required. The signal is then converted to 16-bit values, which are used by the digital-to-analog converter on the FPGA to output a voltage ranging from 0–2.5 V. This voltage is converted to a current using a custom-built voltage-to-current converter, which has the measured transimpedance gain

$$\mathrm{VtoI}(V_{in}, G) = G \times (-0.3224 V_{in} + 0.04082V)/999.001\Omega, \quad (13)$$

where we use $G = 2.5$. In total, we generate 5 different random signals used to collect 5 different datasets for training and model evaluation.

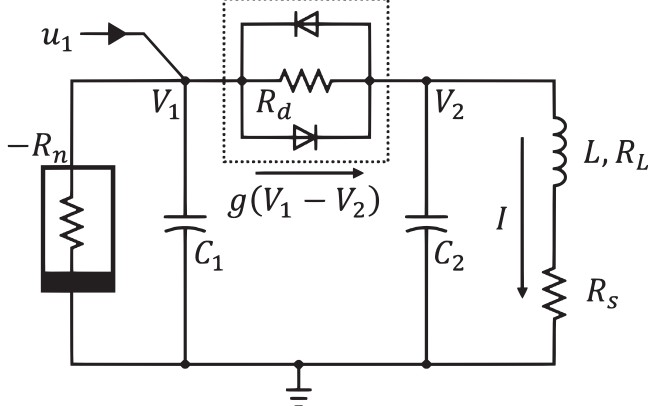

**Fig. 4 | Double scroll electronic circuit.** $R_n = 3.15$ kΩ, $C_1 = C_2 = 10$ nF, $L = 55$ mH, $R_L = 355$ Ω, $R_s = 100$ Ω, $R_m = R_L + R_s = 355$ Ω, $R_d = 7.86$ kΩ, the dotted box encloses the nonlinear coupling $g$.

The ridge parameter for each of the 5 datasets is chosen from an array of 100 logarithmically spaced values from $10^{-10}$ to 10 to minimize the prediction RMSE on the validation set. The same datasets are used to train the one-step ahead and two-step ahead controllers. The learned weights and the ridge regression parameters found for the different datasets and controllers are shown in Supplementary Note 5.

## Generating the desired signals

The desired signals are of finite length as they are stored in the FPGA memory. They are repeated every 21 ms. The desired signal for the origin task is a constant $V_{1,des} = 0$. The desired signal for the two USS task is generated using $V_{1,des\,i} = V_{1ss}' \tanh(\sin(\omega \times i)/b)$, where $\omega = 0.0076$, $b = 0.11$ determines how fast the transition is between the maximum and minimum value, with smaller values being faster, and $V_{1ss}' = 0.571$ is the magnitude of the nonzero fixed-points. This value is 2% smaller than the theoretically predicted value, which was found by adjusting the desired signal until the control current was approximately zero at the location of the fixed-points. The random waveform control task is generated similarly to the training perturbation, but now the low value is between $\pm 1.0$, and we use a 20th-order Butterworth low-pass-filter with a cut-off frequency of 2.5 kHz. There is also a smoothing envelope applied to the noise so it can be repeated without any discontinuities. The desired signals are represented using a fixed-point representation, which is described later.

## Control performance characterization

The control performance is characterized using RMSE in Eq. (10), but $x_i$ is the desired signal, and $\hat{x}_i$ is the state of the controlled system. The system is controlled once for a specific task using each of the 5 sets of learned weights. The control is turned on after time $t_0 = 7.5$ ms and is turned off after $t_f = 80$ ms. The RMSE of the control is calculated from time $t_1 = 10$ ms to $t_f$, not including the transient period. The control for each task and set of weights is repeated for values of $K$ ranging from $-2$ to 1 for the linear controller and $\pm 1$ for the NG-RC-based controllers. In the figures and tables that characterize the RMSE, the error bars and uncertainties are represented by the standard error, which is given by the standard deviation / $\sqrt{\text{num trials}}$, where num trials is 5. The magnitude of the control perturbations is characterized by the root mean square of the signal, or

$$\text{RMS} = \sqrt{\langle u_1^2 \rangle}, \tag{14}$$

which is calculated over the same time interval as the RMSE and uncertainties are reported using the standard error metric.

## Power consumption and energy uncertainty

The uncertainty of all stated power consumption and energy is accounted for in the precision of the Rigol DM3068 Digital Multimeter used to measure the current and the voltage of the MPJA DC power supply supplied to the FPGA device.

## Fixed-point arithmetic

The NG-RC and control law are evaluated using fixed-point arithmetic. All numbers are constrained to 18-bits whenever possible, as the FPGA on-chip only support up to $18 \times 18$ bit multiplications. Multiplying numbers with higher precision requires more than one of these multipliers, which would utilize more resources on the FPGA, slow down the computation, and consume more energy.

To describe the fixed-point representations, we use the Qm.n format, where Q represents the sign bit, m is the number of integer bits, and n is the number of fractional bits. All the features are represented in the Q0.17 form and the weights are also represented using 18-bits, but their fixed-point representation is adjusted based on the scale of the weights which is affected by the ridge parameter.

The representation used for the weights is also used for the feedback gain $K$, but $\hat{W}_u^{-1}$ uses a different representation as the numbers typically require a larger range. The desired signal has the same representation as the features. The different representations of the weights used in the different trials can be seen in Supplementary Note 5.

Quantities that have inherently less precision that 18-bits, such as $V_1$ and $V_2$ measured using 12-bit analog to digital converters, and $u_1$ which is output using a 16-bit digital to analog converter, have the least significant bits padded with 0 s to achieve the Q0.17 form.

## System identification

To select the polynomial features used in the NG-RC, we use a regularized version of the routines found in SysIdentPy[28]. We provide SysIdentPy with $V_{1,i}$, $V_{2,i}$, $u_{1,i}$ and $V_{1,i} - V_{2,i}$ from an example training dataset, and the model is trained to predict $V_{1,i+1}$. The difference of the voltages is added to the feature dictionary because it appears in the differential equations for the system, but in practice any sum or differences of the system variables may be used.

We use the forward regression orthogonal least squares (FROLS) model, with the nonlinear autoregressive moving average model with exogenous inputs (NARMAX) option. We modify the source code for the estimator to include ridge regression (see Eq. (10)) to avoid overfitting and improve model stability[29]. We choose the maximum polynomial degree as 3, and we chose the maximum number of time delays to be 2, as these parameters have been found to give good performance[15]. We then use SysIdentPy to calculate the information criteria of all the possible polynomial combinations for those parameters, and limit the number of terms in the information criteria to 30. We choose a relatively large ridge parameter of $10^{-2}$ as we find it causes the minimum information criteria to occur when less terms are included. The result of this feature selection is shown in Fig. 5.

SysIdentPy finds the minimum information criteria occurs when 14 terms are included in the polynomial, but because the information criteria remains relatively flat after the first 9 terms, we select only these to reduce the model complexity. We also include $u_{1,i-1}$ as we found it improves the control. We do not use the weights learned by SysIdentpy, instead we manually code these polynomial features into our custom Python code that trains and validates the model, which is then transferred to the FPGA.

## Software and devices

The FPGA is compiled and resources are estimated using Quartus Prime 21.1, the training, validation, and analysis of the controlled system is done in Python 3.9.0, using Numpy 1.24.3, Scipy 1.11.1, and the fixed-point calculations are replicated using fxpmath[38] 0.4.8 on an x86-64 CPU running Windows 11. The FPGA is the Max 10 10M50DAF484C6G on a Terasic Max 10 Plus development board.

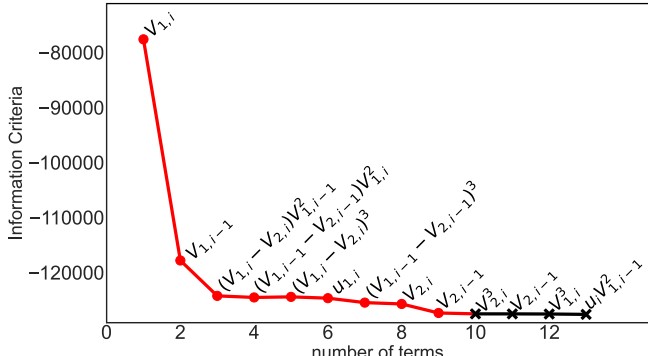

**Fig. 5 | The information criteria of features calculated by SysIdentPy.** More negative means less information is contained in the features. The red circles are the features used in the model, and the black xs are excluded.

## Data availability

The data used to create each figure in this manuscript and the Supplementary Material are available in figshare with the identifier "https://doi.org/10.6084/m9.figshare.25534552"[39].

## Code availability

The Python and FPGA code with example data are available in figshare with the identifier "https://doi.org/10.6084/m9.figshare.25534621"[40].

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

## Acknowledgements

We gratefully acknowledge the financial support of the U.S. Air Force Office of Scientific Research, Contract #FA9550-22-1-0203.

## Author contributions

R. K. and W. A. S. B. performed the experimental studies and analysis. D. J. G. developed the basic control algorithm, interpreted the results, and supervised the work. All authors prepared the manuscript.

## Competing interests

Daniel Gauthier is a co-founder of ResCon Technologies, LLC, which is commercializing the application of reservoir computing. The remaining authors declare no competing interests.
