## [Peer Review File · Nature Communications]

REVIEWER COMMENTS

Reviewer #1 (Remarks to the Author):

The paper about „Controlling Chaos Using Edge Computing Hardware” which is written by Kent et al. , presents an effective control algorithm that is implemented into an FPGA architecture. The authors show that the algorithm is able to control different envisioned dynamics, including chaotic evolution, by using the example of a double scroll circuit. Especially the control of the chaotic dynamics is something that is very hard to reach.

The authors have a long standing record for the topic of control of dynamical systems. In this paper they apply a machine learning based algorithm (next-generation reservoir computing) that is first trained to learn the envisioned dynamics and then used to stabilize it. There are many real world problem where this control strategy might be applicable, though a real example is not given. I think the paper is very interesting, well written, and the results should be useful for a broad range of people dealing with control problems. To put the values of the results into the context of existing results, some more comparison between other methods (linear control, Pyragas control etc..) would be helpful to convince the reader of the advantage gained here.

Also, the authors state that „The closest previous work is by Canaday et al.²³ who controlled the same chaotic circuit using an inverse-based algorithm“. While this citation refers to a PhD thesis, there is another publication by the authors and Canaday (which is not cited) that discusses “Model-free control of dynamical systems with deep reservoir computing“. It presents results on the invers control method for the Lorenz system. I feel that a bit more discussion is needed here to show the new aspects of the presented work (especially because some of the figures look very similar at a first glance).

In its present form I cannot suggest publication in Nature Communication and I think revisions are needed. Specifically, my questions and comments that need to be addressed by the authors are:

Questions/Comments:

1) In the discussion the authors mention that the setup is robust to noise, while they mention in line 112 that it is susceptible to latencies. Can that be specified? How stable is the algorithm if the system under control changes (drifts) between training and control? How much latency can be present before the algorithm fails?

2) In line 182 the authors state “An NG-RC is equivalent to a traditional RC“. Is that a mathematically proven fact? If yes, please cite.

3) In Fig3 the RMS of the control current is given, but it is hard to judge about the absolute value. Is the control signal converged in the regione of good performance (one criticism of the inverse method was that it is not)?

4) In Fig3 the domain of control for the different dynamics is shown. How strong does the domain depend on the feature vector? A comparison of the performance with another feature vector would be very helpful. (as already mentioned above also a comparison to standard control methods would help here)

5) In line 121 the authors state: the advantage of the approach is that it does not need a physical model. However, there are other closed loop schemes that also work without physical models. Are they less reliable?

Typos:

- line119 discreate-time

- line 131 of the of the

Reviewer #2 (Remarks to the Author):

The manuscript focuses on developing a new type of nonlinear controller by utilizing so-called next-generation reservoir computing. The findings are mainly experimental and numerical, with a main claim that the proposed controller design is an energy-efficient framework useful for edge applications. While the main concept of the paper, that is to integrate a lightweight ML component into the control loop, is definitely interesting, I found the paper itself incomplete from several aspects and does not rise to the level of quality to be publishable in Nature Communications.

First, the logical flow of the paper - results part in particular, is not very clear. The control problem which is the core part of the paper, is not even defined. Concepts, notations and methods are presented in an ad hoc fashion, on the fly, making it very difficult to appreciate the fundamental science underlying this work.

Secondly, in terms of the proposed method itself, I am surprised that no thorough comparison with existing works are given. In fact, even the idea of using RC for control is not new (e.g., the work by Canada et. al., which is mentioned in the paper but not carefully analyzed), not to say alternative classical nonlinear control designs. Thus it is not convincing, based on the content of the current manuscript, that NGRC integrated into a controller as proposed serves as a better controller - regardless of other considerations such as power consumption and size.

Third, the proposed control approach of injecting a perturbation to learn the dynamics and design corresponding control laws is in fact a common practice in the field of adaptive control. For instance, in the standard framework of extreme seeking control/dither learning control, the perturbation therein is usually called a dither. The authors should consider a more comprehensive literature study in order to fully support the claim of novelty.

Finally, in a more general perspective, studying control design for edge devices and edge computing is not a new research problem. What is the particular reason that "controlling chaos" on an edge device is chosen to be the only types of demos shown in the paper? Why not general robotics motion, for instance?

Reviewer #3 (Remarks to the Author):

In this paper, the authors proposed a feedback controller based on next-generation reservoir computing to control a chaotic system to an arbitrary time-dependent state. The controller was implemented on a field-programmable gate array. The authors regarded this problem of controlling a chaotic system "the hardest-of-the-hard control problems" or "a known difficult nonlinear control problem". Experimental results of controlling a chaotic circuit about some unstable steady states and a random orbit were presented. A highlight of the work, as the authors claimed, is that the energy required for control is "38 nJ per model evaluation, well below other algorithms, even without a systematic power optimization."

The current version of the manuscript is not suitable for Nature Communications.

Major:

1. Controlling chaos using arbitrarily small control signals or perturbation based on measured time series, without any knowledge about the equations of the system to be controlled, has been demonstrated for a large variety of systems in different fields of science and engineering for 33 years since the pioneering work of Ott, Grebogi, and Yorke (OGY) in 1990. In fact, stabilizing unstable steady states and unstable periodic orbits using small perturbations has been amply demonstrated in many different types of experimental platforms ranging from chaotic circuits and chaotic chemical reactions to chaotic cardiac and brain systems. Extension of the OGY method to controlling arbitrarily chaotic trajectories was also proposed and experimentally realized about 30 years ago. In the previous works, the required energy can be arbitrarily low, depending on the length of the chaotic transient preceding control realization. Presenting a number such as "38 nJ" is not very meaningful. While the authors compared this energy value with the energies required by three other controllers: low-power PID control, autonomous driving, and aircraft control, a direct comparison with the OGY control method is missing. Such a comparison is necessary, as OGY control is the leading method of controlling chaos in the field. More specifically, for OGY control there is a scaling relation between the required control energy and the transient time. The authors should conduct experiments on their chaotic Chua's circuit (1) by using the OGY method to obtain the energy scaling and (2) by using their proposed machine-learning controller to obtain this scaling. The two scaling relations can then be compared - only when such a systematic comparison has been performed can any claim of "38 nJ" or requiring less energy than other algorithms be justified or meaningful.

2. It is misleading to state that the problem of stabilizing a chaotic system around some desired orbit (e.g., an unstable steady state, an unstable periodic orbit, or even a chaotic orbit) is "the hardest-of-the-hard control problems" or "a known difficult nonlinear control problem". If the problem was solved more than three decades ago and has been demonstrated in an extremely diverse array of experimental platforms, it may be inappropriate to call it "hard."

3. What the authors demonstrated was to control their chaotic circuit to one of the two unstable steady states or a chaotic orbit, with main discussion focused on controlling the unstable steady states. More examples should be presented, especially examples of successful control of a number of unstable periodic orbits of slightly higher periods, such as period-2 or period-3. Control flexibility in switching to control different unstable periodic orbits should also be demonstrated.

4. The double-scroll circuit is known to be a low-dimensional chaotic system with only one positive Lyapunov exponent. In the past, controlling high-dimensional chaotic systems with two or more positive Lyapunov exponents was demonstrated. To convince the readers that the authors' reservoir-computing controller is effective or useful, successful control of such high-dimensional chaotic systems should be demonstrated.

Minor:

1. For the number "38 nJ", what is the error bar?

2. In Supplementary Table 5, the required control energy values are listed as numbers with two digits after the decimal point.

What are the uncertainties or error bars of those numbers?

3. A number of figures (e.g., Figs. 3 and 5 in the main text) have low resolution and are of poor quality.

4. Introduction should be rewritten to account for the proper history of the field of chaos control, widely regarded as starting from the 1990 OGY work. The small energy requirement for OGY control should be emphasized. Previous experimental control of chaos based on the OGY principle (e.g., the many works of Ditto et al.) should also be included to place the present work in a proper and historically correct context.

Response to reviewers:

Dear Editor,

We thank the reviewers for their detailed comments and suggestions, which has greatly improved our presentation. We have thoroughly addressed their criticisms, and our work is now ready for publication. In our resubmission, we provide two versions of the manuscript—one with the corrections in red text, and the other without red text. Below, we give a point-by-point response to the referees' comments.

Robert Kent, on behalf of all the authors.

Reviewer 1:

1.1: The paper about “Controlling Chaos Using Edge Computing Hardware” which is written by Kent *et al.*, presents an effective control algorithm that is implemented into an FPGA architecture. The authors show that the algorithm is able to control different envisioned dynamics, including chaotic evolution, by using the example of a double scroll circuit. Especially the control of the chaotic dynamics is something that is very hard to reach. The authors have a long standing record for the topic of control of dynamical systems. In this paper they apply a machine learning based algorithm (next-generation reservoir computing) that is first trained to learn the envisioned dynamics and then used to stabilize it.

We thank the referee for the positive comments and look forward to thoroughly addressing all of their comments.

1.2: There are many real world problem where this control strategy might be applicable, though a real example is not given.

The reviewer states that a real example is not given; however, we argue that controlling a chaotic circuit serves as an apt real-world example for several reasons. First, developing a controller for a system that exhibits chaos introduces additional challenges when compared to a typical linear or nonlinear control problem because chaotic dynamics can be more complex and unpredictable with a finite correlation time and hence control-loop latency plays a more important role. Because our algorithm is successful in controlling a real-world device in this more difficult class of systems, we argue it should succeed in the more typical examples the reviewer envisions.

Second, many real-world examples have dynamics that evolve more slowly with a time scale much longer than the control-loop latency of the FPGA-based controller. For instance, controlling a robotic arm, self-driving automobiles, or quad-copter have dynamics that evolve on the second to 100's of millisecond timescale, whereas our system evolves on the sub-millisecond timescale. This timescale of the chaotic circuit is well matched to the sampling rate of the edge computing device used in our study; it is straightforward to use the same device at

any slower timescale that matches the real-world systems mentioned above or to use a higher-end device that operates on the nanosecond timescale, such as the Xilinx ZCU216. Third, we chose this problem to clearly illustrate the advantages over the previous approach that uses a traditional reservoir computer to control the same system by Canaday *et al.*¹⁴ Finally, we now clearly point out in the manuscript the failure of previous chaos control techniques as discussed in detail below.

Applying our control technique to other real-world examples is certainly interesting and is part of our ongoing research, but we believe the work presented in the manuscript is a hard enough first example to stand on its own.

1.3: I think the paper is very interesting, well written, and the results should be useful for a broad range of people dealing with control problems.

We thank the reviewer again for the positive comments.

1.4: To put the values of the results into the context of existing results, some more comparison between other methods (linear control, Pyragas control etc..) would be helpful to convince the reader of the advantage gained here.

We agree with the reviewer's suggestion to include comparisons with other control techniques such as linear control and the Pyragas time-delay control. To compare our approach with linear control, we show that our algorithm reduces to a linear controller by replacing a single term, as described in the second to last paragraph in the Control Law section on page 5

“For comparison, we compare our nonlinear controller to a simple linear proportional feedback controller. We can make a simple adjustment to the control law described above to realize a linear controller. Consider the case when the sampling time Δt is short relative to the timescale of the system's dynamics, we can approximate the flow $\hat{\mathbf{F}}_{i+m}$ in the control law (3) with the current state \mathbf{Y}_i to arrive at a linear control law. This is motivated by approximating the flow using Euler integration with a single step $\mathbf{Y}_{i+1} \approx \mathbf{Y}_i + \Delta t \mathbf{f}(t_i, \mathbf{X}_i)$, where $\mathbf{f}(t_i, \mathbf{X}_i)$ is the vector field governing the dynamics, and observing that $\mathbf{Y}_{i+1} \approx \mathbf{Y}_i$ when Δt approaches zero.”

We compare the performance of this linear control algorithm and the full ML control law with one and two-step ahead predictions by testing them on the same control tasks and computing the same error metrics. We include a paragraph that summarizes these error metrics at the start of the results section on page 6

“In this section, we present an overview of the control tasks and examine example trajectories of the system before and after control begins. Then, we look at the control error and size of the control perturbations as a function of the feedback control gain K for the one-step ahead controller. Last, we compare the minimum errors for the linear, one-step, and two-step ahead controllers.”

We also include Table 1, which shows the minimum error for each task and controller type, and we provide additional paragraphs at the end of the results on page 8 highlighting the key findings

“The minimum error for each task and controller type is given in Table 1. We see that the NG-RC controllers outperforms the linear controller on every task, except when the system is stabilized at the nonzero USSs because the dynamics are approximately linear there.

The advantage of the NG-RC controller is evident for the random waveform task, where we observe a 1.7× reduction in the error compared to the linear controller. This task is especially challenging for the linear controller because the desired state visits a wide range of phase space with different local linear dynamics and the sampling rate is too slow for $\mathbf{Y}_{i+1} \approx \mathbf{Y}_i$ to be a good approximation, making the controller ineffective at cancelling the nonlinear dynamics of the system. Additional performance metrics, such as the error, size of control perturbations, and sensitivity to control-loop latency are given in the Supplementary Note 3.”

We include Supplementary Note 3: Additional performance metrics, in which we compare the error, the size of the control perturbations, and the sensitivity to control loop latency. Note that we train the NG-RC controller using the minimum control loop latency, and increase the latency during the control phase without retraining the model. The key points of this section are listed here

- The linear and NG-RC controller with two-step ahead prediction have a broad range of feedback gain K over which control is stable (domain of control) for most tasks, but the linear controller has a small domain of control for the random waveform task.
- The linear controller requires large control perturbations for the random waveform task compared to the NG-RC controller.
- The linear controller has the largest domain of control for the origin and two USS task as the control loop latency is increased, but fails to maintain stability on the random waveform task, in contrast to the two-step ahead controller that maintains stability for this task with up to 4 μ s of added delay.

These additional results are shown in Supplementary Figure 2, shown below

We also include Supplementary Table 1, which gives the size of the control perturbations that correspond to the lowest control error for each controller type and task.

In addition to performance comparisons, we include the power consumption of the linear controller in the second paragraph of the Control Resources section on page 10

“In contrast, the linear controller requires only 7.5 ± 7.0 nJ per inference, which is primarily due to the decreased number of multiplications.”

and the FPGA resources of the linear controller in the third paragraph of the Control Resources section on page 10

“In contrast, the linear controller uses 986 (2%) logic elements, 604 registers (1.2%), and 4 multipliers.”

To address the request for comparison with Pyragas control,⁵ we note that it is impossible to use this approach to control the chaotic circuit to most of the states we study here. For example, Chang *et al.*¹⁶ use an extension of the Pyragas control scheme known as extended time-delay auto synchronization (ETDAS),³¹ but fail to control the system to the origin of the system, stating “it falls into the class of uncontrollable states.” To indicate this previous result in the text, we include the following to the second paragraph of the results section on page 6

“We apply our nonlinear control algorithm with three challenging tasks. The first is to control the system to the unstable **steady state (USS)** located at the origin [$V_1 = 0$, $V_2 = 0$, $I = 0$] of phase space starting from a random point located on the strange attractor. **Previous work by Chang *et al.*¹⁶ fail to stabilize the system at the origin using an extended version of the Pyragas approach.³¹”**

Chang *et al.*¹⁶ are successful in stabilizing either of the unstable steady states (USS) at the center of the scrolls, but this approach cannot be used to rapidly switch between these USSs. The primary reason is that this approach stabilizes the system at the first nonzero USS if $V_1 < 0$, and the second USS if $V_1 > 0$. Therefore, switching from one USS to the other requires the controller to stop applying control perturbations, wait until the system naturally crosses the line $V_1 = 0$, and then resume applying control perturbations. This process cannot happen within the requested transition time. To explain this, we include the following to the to the third paragraph in the results section on pages 6-7

“The second task is to control the system about one of the **USSs** in the middle of one of the “scrolls” and then rapidly guide the system to the unstable steady state on the other scroll and then the other in rapid succession. **Rapid switching between these USSs is unattainable using Pyragas-like approaches because the controller must wait for the system to cross into the opposite basin of attraction defined by the line $V_1 = 0$,¹⁶ which cannot occur within the requested transition time.**”

Last, Pyragas-like control schemes can only control systems to unstable periodic orbits or unstable fixed-points, making the control to a random waveform impossible. To make this clear to the reader, we include the following to the third paragraph in the introduction on pages 1-2

“A particular difficult control task is to stabilize a nonlinear system displaying a complex behavior known as chaos, which is characterized by sensitivity to initial conditions and tiny perturbations. Previous approaches, such as that pioneered by Ott, Grebogi, and Yorke (OGY)⁴ and by Pyragas,⁵ are restricted to stabilizing unstable states embedded in the dynamics and are restricted to making small adjustments to the accessible system parameters and hence control can only be turned on once the system is close to a desired state.”

Additionally, we include the following to the fourth paragraph in the results section on page 7

“Last, we control the system to a random trajectory, which determines if the system can be controlled to an arbitrary time-dependent state. This task is difficult because the controller must cancel the nonlinear dynamics at all points in phase space, requiring an accurate model of the flow. This task is impossible for Pyragas-like approaches, as they are limited to controlling the system to UPOs and USSs.”

In response to later comments, we also discuss why the other classic method of chaos control by Ott, Grebogi, and Yorke (OGY)⁴ fail to control to these trajectories as well.

1.5: Also, the authors state that „The closest previous work is by Canaday *et al.*²³ who controlled the same chaotic circuit using an inverse-based algorithm“. While this citation refers to a PhD thesis, there is another publication by the authors and Canaday (which is not cited) that discusses “Model-free control of dynamical systems with deep reservoir computing”.

We agree with the reviewer, and we have replaced the citation to Canaday’s thesis with the recommended citation throughout the text.

1.6: It presents results on the inverse control method for the Lorenz system. I feel that a bit more discussion is needed here to show the new aspects of the presented work (especially because some of the figures look very similar at a first glance).

We agree with the reviewer, which has prompted us to include additional comparisons with the work by Canaday *et al.*¹⁴ First, we explain the novelty of our approach and the aspects in which we improve upon the work by Canaday *et al.*¹⁴ in paragraphs 5-7 in the introduction

“One example approach is a reservoir computer^{12,13} (RC), a best-in-class ML algorithm for learning dynamical systems. The previous work by Canaday *et al.*¹⁴ uses a RC to learn an inverse-based control algorithm, but fails to achieve accurate control. We believe these shortcomings are due to the learned inverse not being unique or due to control-loop latency that was not learned.

To avoid these problems, we use a feedback linearization algorithm combined with a highly efficient next-generation reservoir computer (NG-RC)¹⁵ which greatly simplifies the controller and reduces the computational resources. Here the NG-RC predicts the future state of the system, which is feedback via the controller to cancel the nonlinear terms of the systems nonlinear dynamical evolution, while simultaneously providing linear feedback to stabilize the system to the desired time-dependent state.

We demonstrate that a single NG-RC-based controller can stabilize a chaotic electronic circuit¹⁶ to trajectories that are impossible for the classic chaos control methods and has lower error than the inverse-control algorithm used by Canaday *et al.*¹⁴ as well as a linear controller.”

We also include additional details explaining the differences between a traditional reservoir computer (RC) and a next-generation reservoir computer (NG-RC) in the last paragraph of the Reservoir Computing section on page 6

“Previous studies have successfully used an RC to control a robotic arm²⁶ and a chaotic circuit,¹⁴ for example, using an inverse-control method.²⁷

The next-generation reservoir computing algorithm has similar characteristics to a traditional RC,²⁸ but has fewer parameters to optimize, requires less training data, and is less computationally expensive.^{15”}

The most significant change we make regarding this point is to include a comparison of the computational complexity between our approach and that of Canaday *et al.*¹⁴ in Supplementary Note 1. Here, we compare the number of multiplications, evaluations of tanh, and trainable parameters required to implement each model. We list the main points below

- For a one-layer controller, Canaday *et al.* require 270 multiplications and 30 evaluations of tanh to evaluate the control law, and have 30 trainable parameters. These numbers double for a two-layer controller which has closer (but still worse) performance to our approach.
- Our NG-RC-based controller requires 18 multiplications per evaluation of the control law, and does not require any evaluations of tanh, using fewer look up tables. We only require 10 trainable parameters.

We summarize these results, and include additional comparisons such as the amount of data needed to train the controller in the main text in the second paragraph of the discussion on page 10

“When compared to a linear controller, our approach achieves a 1.7× smaller error on the most difficult control task, requiring only a small increase in the percentage of available FPGA resources and the power consumption. Furthermore, we achieve better performance than the RC-based inverse control law approach by Canaday *et al.*¹⁴ while using 4× less training data, 6× fewer trainable parameters, 30× fewer multiplications, and 60 fewer evaluations of the hyperbolic tangent function. Additionally, we use 18-bit arithmetic rather than 32-bit arithmetic, which uses fewer resources on the FPGA device. These differences demonstrate that a nonlinear controller based on the NG-RC algorithm achieves higher performance using substantially few compute resources.”

We believe these additional comparisons to the work by Canaday *et al.*¹⁴ make the benefits of our approach clear to the reader.

In its present form I cannot suggest publication in Nature Communication and I think revisions are needed. Specifically, my questions and comments that need to be addressed by the authors are:

Questions/Comments:

1.7: 1) In the discussion the authors mention that the setup is robust to noise, while they mention in line 112 that it is susceptible to latencies. Can that be specified? How stable is the algorithm if the system under control changes (drifts) between training and control? How much latency can be present before the algorithm fails?

We have answered these questions in our response to comment 1.4 and in our inclusion of Supplementary Figure 2.

1.8: 2) In line 182 the authors state “An NG-RC is equivalent to a traditional RC”. Is that a mathematically proven fact? If yes, please cite.

This statement is not a mathematically proven fact, but there is an existing work by Bollt²⁸ that highlights their similarities. Bollt shows that an RC with linear activation and a readout consisting of a sum of nonlinear functionals is equivalent to a nonlinear vector autoregression (NVAR) machine, which is the basic principle of the next-generation reservoir computer.¹⁵ We have corrected this which can be seen in our response to comment 1.6.

1.9: 3) In Fig3 the RMS of the control current is given, but it is hard to judge about the absolute value. Is the control signal converged in the region of good performance (one criticism of the inverse method was that it is not)?

To see that the control signal has converged in the region of good performance, we provide additional plots that show the maximum absolute value of the control current shortly after the control is turned on (2.5ms), which is shown for each controller type in Supplementary Figure 3.

We see that the maximum control current is small at the USSs, but large when steering between the two USSs and the random waveform, because the trajectories contain rapid changes which require large perturbations.

The RMS of the control current is still the best metric for evaluating the convergence of the system, as it should be very small when the system is stabilized at a USS. To highlight this point, we include the RMS of the current when the system is stabilized at either nonzero USS (purple diamonds) to Supplementary Figure 2 (see response to comment 1.4) and Figure 3, shown below

Figure 3. Performance of the one-step ahead NG-RC controller. (a) RMSE of the control and (b) RMS of the control current when controlling the system to the phase-space origin (red circles), back-and-forth between two USSs (green triangles), to a random waveform (blue squares), and stabilized at either nonzero USS (purple diamonds) as a function of the control gain K .

We see that the control current required to stabilize the system at the nonzero USSs is very small, confirming that the control signal is converged.

The reviewer states that in the approach by Canaday *et al.*,¹⁴ the control signal is not converged when the performance is good. This is evident by a large RMS current when the system is at USSs, and additional controller layers increase this RMS further. Our controller does not suffer from this issue.

1.10: 4) In Fig3 the domain of control for the different dynamics is shown. How strong does the domain depend on the feature vector? A comparison of the performance with another feature vector would be very helpful. (as already mentioned above also a comparison to standard control methods would help here)

The feature vector is chosen using system identification techniques, as described in the final paragraph of the Reservoir Computing section on page 6

“We perform system identification^{29,30} to select the most important components of the model.”

The features found using these techniques are optimal for this system with the given parameters and sampling rate, thus we believe it is not necessary to show how the domain of control depends on the elements of the feature vector. However, we have compared the domain of control with a linear controller, which can be seen in our response to comment 1.4.

As for other standard control methods, such as Pyragas control,⁵ we have explained that this approach is not capable of controlling the system to any of the trajectories we study here, so the domain of control cannot be compared.

1.11: 5) In line 121 the authors state: the advantage of the approach is that it does not need a physical model. However, there are other closed loop schemes that also work without physical models. Are they less reliable?

The two most well-known methods in chaos control, OGY⁴ and Pyragas,⁵ do not require a physical model. In the OGY approach, the unstable periodic orbits (UPOs) are learned through experimental observations, and the state of the system is measured, and feedback is applied that is proportional to the difference between the current state of the system and UPO. This approach is limited to small adjustments to the accessible system parameters. In contrast, Pyragas control requires only the knowledge of the period of the desired unstable periodic orbit. Our approach is different from these classic control methods because it learns a physical model of the system, allowing it to make predictions of the future dynamics to perform feedback linearization. The benefit is that our approach learns to cancel the nonlinear dynamics of the system anywhere on the attractor, allowing it to control the system to arbitrary trajectories, whereas OGY and Pyragas are limited to UPOs and USSs.

We are aware that the OGY approach has been extended by Shinbrot *et al.*³² to steer chaotic systems to arbitrary states in the attractor, but this approach still cannot control the chaotic circuit to all the trajectories we study here, which is discussed in detail later.

1.12: Typos:

- line119 discreate-time

We thank the reviewer for pointing out this mistake and have made the requested change in the Control Law section in the second paragraph of the Control Law section on page 5

“discrete-time systems”

- line 131 of the of the

Again, we thank the reviewer for pointing out this mistake and have made the requested change in the Control Law section in the second paragraph on page 4

“of the system”

Reviewer 2:

2.1: The manuscript focuses on developing a new type of nonlinear controller by utilizing so-called next-generation reservoir computing. The findings are mainly experimental and numerical, with a main claim that the proposed controller design is an energy-efficient framework useful for edge applications. While the main concept of the paper, that is to integrate a lightweight ML component into the control loop, is definitely interesting, I found the paper itself incomplete from several aspects and does not rise to the level of quality to be publishable in Nature Communications.

We appreciate that the reviewer found our work interesting, and we hope to thoroughly address all their concerns.

2.2: First, the logical flow of the paper - results part in particular, is not very clear.

We appreciate the reviewer for bringing this to our attention. To address this, we have added a paragraph at the beginning of the Results section on page 6, serving as a roadmap for the content that follows, which can be seen in our response to comment 1.4.

In addition, we have divided the Results section into the following subsections: Control Tasks, Example Trajectories, and Control Performance. In the Control Tasks subsection, details have been added to explain the difficulty of each task, and why they cause the classic chaos control methods to fail, which connects the results section to the claims made throughout the paper. In the Control Performance section, we have included comparisons with a linear controller (see response to comment 1.4), which places our results in the proper context and make the benefits of our approach clear to the reader.

2.3: The control problem which is the core part of the paper, is not even defined.

We have stated the control problem loosely in the Introduction

“A controller automatically adjusts accessible system parameters based on real-time measurement of the system to guide it to a desired state.”

and in the System Overview section

“The goal of the controller is to guide the system to the desired state by perturbing its dynamics based on a nonlinear control law.”

but we have included additional text at the beginning of the Control Law section on page 4 to define it more formally

“We formalize the control problem described in the previous sections as follows: the accessible variables of the chaotic circuit are denoted by $\mathbf{X} \in \mathbb{R}^{d'}$ are the inputs to a nonlinear control law that specifies the control perturbations $\mathbf{u} \in \mathbb{R}^d$ ($d \leq d'$) necessary to guide a subset of the accessible system variables $\mathbf{Y} \in \mathbb{R}^d$ to a desired state $\mathbf{Y}_{des} \in \mathbb{R}^d$.”

We want to emphasize that this work is intended for a broad audience, so a more formal mathematical definition of the control problem is to be avoided.

2.4: Concepts, notations and methods are presented in an ad hoc fashion, on the fly, making it very difficult to appreciate the fundamental science underlying this work.

We would like to highlight that the first reviewer said the paper is well written and useful for a broad range of people dealing with control problems (see comment 1.3). We have restructured the Control Law section and simplified the notation in the Control Law section and the Methods to make the presentation more clear. We think the concepts, notations, and methods are presented in a logical way, but we would appreciate some specific suggestions on how to best improve the presentation for a general audience.

2.5: Secondly, in terms of the proposed method itself, I am surprised that no thorough comparison with existing works are given. In fact, even the idea of using RC for control is not new (e.g., the work by Canaday et al., which is mentioned in the paper but not carefully analyzed), not to say alternative classical nonlinear control designs.

We have addressed these points in responses to previous comments (1.4 and 1.6). We go into more detail about another classical nonlinear control design, the OGY method,⁴ in response to later comments.

2.6: Thus it is not convincing, based on the content of the current manuscript, that NGRC integrated into a controller as proposed serves as a better controller - regardless of other considerations such as power consumption and size.

We believe the additions we have made to the manuscript make it clear that the NG-RC-based controller has many advantages over previous control techniques and therefore serves as a better controller. First, it is $1.7\times$ more accurate than a linear controller and uses has only a marginal increase in power consumption and resource utilization. Second, it achieves much higher accuracy than the work by Canaday *et al.*¹⁴ using a similar machine learning architecture and requires vastly fewer resources. Third, it can be used to control the system to trajectories that are impossible for traditional chaos control approaches. Fourth, it does not require a computationally expensive and slow optimization at every timestep to compute the optimal control action like in model-predictive control. Last, we believe the minimal power consumption and size are crucial to future low-power edge computing applications.

2.7: Third, the proposed control approach of injecting a perforation to learn the dynamics and design corresponding control laws is in fact a common practice in the field of adaptive control. For instance, in the standard framework of extreme seeking control/dither learning control, the perturbation therein is usually called a dither. The authors should consider a more comprehensive literature study in order to fully support the claim of novelty.

We thank the reviewer for informing us on this topic. To address this, we include the following to the second to last paragraph in the System Overview section on page 4,

“The control current perturbs the chaotic dynamics so we can learn how it responds to perturbations at many locations in phase space, **which is common in extremum-seeking control,²⁰ for example.**”

where reference 20 is *Real Time Optimization by Extremum Seeking Control* by Ariyur and Krstic.

2.8: Finally, in a more general perspective, studying control design for edge devices and edge computing is not a new research problem. What is the particular reason that “controlling chaos” on an edge device is chosen to be the only types of demos shown in the paper? Why not general robotics motion, for instance?

We addressed these questions in our response to comment 1.2.

Reviewer 3:

3.1: In this paper, the authors proposed a feedback controller based on next-generation reservoir computing to control a chaotic system to an arbitrary time-dependent state. The controller was implemented on a field-programmable gate array. The authors regarded this problem of controlling a chaotic system "the hardest-of-the-hard control problems" or "a known difficult nonlinear control problem". Experimental results of controlling a chaotic circuit about some unstable steady states and a random orbit were presented. A highlight of the work, as the authors claimed, is that the energy required for control is "38 nJ per model evaluation, well below other algorithms, even without a systematic power optimization."

The current version of the manuscript is not suitable for Nature Communications.

We thank the reviewer for their time and consideration, and we hope to address thoroughly address their comments and concerns.

Major:

3.2: 1. Controlling chaos using arbitrarily small control signals or perturbation based on measured time series, without any knowledge about the equations of the system to be controlled, has been demonstrated for a large variety of systems in different fields of science and engineering for 33 years since the pioneering work of Ott, Grebogi, and Yorke (OGY) in

1990. In fact, stabilizing unstable steady states and unstable periodic orbits using small perturbations has been amply demonstrated in many different types of experimental platforms ranging from chaotic circuits and chaotic chemical reactions to chaotic cardiac and brain systems. Extension of the OGY method to controlling arbitrarily chaotic trajectories was also proposed and experimentally realized about 30 years ago.

The reviewer points out that the OGY method⁴ has been extended to control arbitrary chaotic trajectories, which we believe refers to the work by Shinbrot *et al.*³², which has been experimentally realized in “Using the Sensitive Dependence of Chaos (the “Butterfly Effect”) to Direct Trajectories in an Experimental Chaotic System” by Shinbrot *et al.* (1992), among others. This approach can be used to target arbitrary fixed-points present in the dynamics of the system, but the OGY approach and its descendants are still restricted to small perturbations. This means this approach necessarily fails at two control tasks presented in our work: the rapid switching between the nonzero USSs, and the random waveform, as they both require large perturbations. Furthermore, even if the transition time between the two USSs is increased to allow for smaller perturbations, the system never visits the vicinity of these two USSs,¹⁶ so the extensions of OGY cannot target these states. Another major drawback of the extended approach is that it can require a large memory to store trees containing possible paths to the desired state,³³ is more susceptible to modeling errors (as found in “Controlling Chaos in a Pendulum Subjected to Feedforward and Feedback Control” by Yagasaki and Uozumi 1997), and the reduced convergence time hinges on the presence of chaotic dynamics. The need for additional memory, particularly in a device such as an FPGA, comes with a large energy cost.

3.3: In the previous works, the required energy can be arbitrarily low, depending on the length of the chaotic transient preceding control realization.

We appreciate this insight, but we would like to highlight that when rapidly switching between USSs or following a random waveform, there is not enough time for a chaotic transient period to occur as the system must follow a rapidly changing trajectory.

3.4: Presenting a number such as “38 nJ” is not very meaningful.

We believe that the power consumption comparisons with the linear controller in response to comment 1.4 places this number in a meaningful context.

3.5: While the authors compared this energy value with the energies required by three other controllers: low-power PID control, autonomous driving, and aircraft control, a direct comparison with the OGY control method is missing. Such a comparison is necessary, as OGY control is the leading method of controlling chaos in the field. More specifically, for OGY control there is a scaling relation between the required control energy and the transient time. The authors should conduct experiments on their chaotic Chua's circuit (1) by using the OGY method to obtain the energy scaling and (2) by using their proposed machine-learning controller to obtain this scaling. The two scaling relations can then be compared - only when

such a systematic comparison has been performed can any claim of "38 nJ" or requiring less energy than other algorithms be justified or meaningful.

As previously stated, we believe this scaling relation is not essential to the discussion because the OGY approach cannot control the system to arbitrary time-dependent states, unlike our approach and other machine learning-based approaches.

Nevertheless, we understand the importance of the OGY method to this discussion, which is why we have made sure to include it in the introduction in response to previous comments, and we include additional text in the results section. Regarding the two USS task, we include the following to the third paragraph in the Results section on page 6

“Additionally, the system never visits the vicinity of these USS¹⁶ and hence large perturbations are required to perform the fast transitions, causing the OGY⁴ approach to fail.”

and regarding the random waveform task, we include the following to the fourth paragraph in the Results section on pages 6-7

“There are extensions of the OGY method that can steer the system to arbitrary points on the attractor,³² but controlling between arbitrary points quickly requires large perturbations, making this approach unsuitable. Furthermore, this approach can require significant memory to store paths to the desired state,³³ increasing the power and resource consumption.”

where reference 32 is Shinbrot *et al.* (1990) and reference 33 is Kostelich *et al.* (1993).

3.6: 2. It is misleading to state that the problem of stabilizing a chaotic system around some desired orbit (e.g., an unstable steady state, an unstable periodic orbit, or even a chaotic orbit) is "the hardest-of-the-hard control problems" or "a known difficult nonlinear control problem". If the problem was solved more than three decades ago and has been demonstrated in an extremely diverse array of experimental platforms, it may be inappropriate to call it "hard."

We have thoroughly explained what makes each of these states difficult or impossible for the classic chaos control methods in response to previous comments. Nevertheless, we make a few changes to the manuscript to indicate that these problems are difficult but not the hardest-of-the-hard control problems. In the abstract, we make the following change

“Here, we show that a nonlinear controller based on next-generation reservoir computing can tackle a difficult control problem: controlling a chaotic system to an arbitrary time-dependent state.”

We also replace the original text in the results

“Lastly, we control the system to a random trajectory, the hardest-of-the-hard nonlinear control tasks.”

to the following in the fourth paragraph in the results section on page 6

“Last, we control the system to a random trajectory, which determines if the system can be controlled to an arbitrary time-dependent state. This task is difficult because the controller must cancel the nonlinear dynamics at all points in phase space, requiring an accurate model of the flow.”

3.7: 3. What the authors demonstrated was to control their chaotic circuit to one of the two unstable steady states or a chaotic orbit, with main discussion focused on controlling the unstable steady states. More examples should be presented, especially examples of successful control of a number of unstable periodic orbits of slightly higher periods, such as period-2 or period-3. Control flexibility in switching to control different unstable periodic orbits should also be demonstrated.

We believe that showing that our approach can control the system to unstable periodic orbits is unnecessary, as the random trajectory is significantly harder than these tasks, and it proves that the system will follow any trajectory provided to it, including UPOs.

In response to the request to demonstrate switching between different UPOs, we point out that we demonstrate rapid switching between two period-1 UPOs, and switching between higher period orbits is still less challenging than the random trajectory, so we believe this is unnecessary.

3.8: 4. The double-scroll circuit is known to be a low-dimensional chaotic system with only one positive Lyapunov exponent. In the past, controlling high-dimensional chaotic systems with two or more positive Lyapunov exponents was demonstrated. To convince the readers that the authors' reservoir-computing controller is effective or useful, successful control of such high-dimensional chaotic systems should be demonstrated.

We are limited to a low-dimensional system because our FPGA device has a limited number of analog-to-digital and digital-to-analog channels, allowing us to measure only two dynamical variables, and feedback to only one. Future work can address this problem by using a device with more input/output channels.

We have shown that we control it to tasks that are difficult for classic chaos control methods despite our system having low-dimensionality and only single Lyapunov exponent, proving the effectiveness of the controller.

Control of higher-dimensional systems with additional positive Lyapunov exponents should be possible using our approach as the success of our controller hinges on the NG-RC learning an accurate model of the system. It has been shown that an NG-RC can learn accurate models of such systems, such as the high-dimensional extended Lorenz 96 system.³⁵

We agree with the reviewer that it would be interesting to apply our control algorithm to higher dimensional systems with more Lyapunov exponents, but this is part of our future work.

Minor:

3.9: 1. For the number "38 nJ", what is the error bar?

Since the previous submission, this number has been reduced to 25.0 ± 7.0 nJ by further optimization of the FPGA code. The uncertainty is calculated using the precision of the volt meter and the current meter.

We have added a statement in the Methods to clarify this

“Power Consumption and Energy Uncertainty

The uncertainty of all stated power consumption and energy is accounted for in the precision of the Rigol DM3068 Digital Multimeter used to measure the current and the voltage of the MPJA DC power supply supplied to the FPGA device.”

3.10: 2. In Supplementary Table 5, the required control energy values are listed as numbers with two digits after the decimal point.

What are the uncertainties or error bars of those numbers?

Supplementary Table 5 has been changed to Supplementary Table 2, and the values have been updated to include uncertainties.

3.11: 3. A number of figures (e.g., Figs. 3 and 5 in the main text) have low resolution and are of poor quality.

The resolution and quality of both figures have been improved. Fig. 5 has been remade for improved readability, which can be seen below

Figure 5. The information criteria of features calculated by SysIdentPy. More negative means less information is contained in the features. **The red circles are the features used in the model, and the black xs are excluded.**

3.12: 4. Introduction should be rewritten to account for the proper history of the field of chaos control, widely regarded as starting from the 1990 OGY work. The small energy requirement for OGY control should be emphasized. Previous experimental control of chaos based on the OGY principle (e.g., the many works of Ditto et al.) should also be included to place the present work in a proper and historically correct context.

We have included citations for the seminal works of OGY and Pyragas in the introduction in response to previous comments and discuss their drawbacks. Given that we do not directly compare our algorithm with these approaches, we believe that additional discussion such as the small energy requirement or many experimental works are not necessary.

Additional changes

Numerous typographic errors have been corrected throughout the text, and small changes have been made to improve clarity, highlighted in red.

We have rearranged the order of the sections and figures in the Supplementary Material.

We have included the author contributions and competing interests statement but we wait to include the data availability and code availability statements until the manuscript is accepted.

REVIEWERS' COMMENTS

Reviewer #1 (Remarks to the Author):

In their revised version the authors carefully addressed all my questions and significantly changed the manuscript. After reading the reports and replies to the other two referees, I also think they answered their issues satisfactorily.

I am convinced that the paper now meets the requirement of novelty and broad interest and I suggest publication in its present form.

Reviewer #2 (Remarks to the Author):

The revised manuscript has addressed all the issues that I have raised in the previous round, and I would recommend publication.

Reviewer #3 (Remarks to the Author):

I appreciate the authors' effort to rebut the referee comments and to revise the paper by addressing some of these comments. The revised paper has been improved to some extent, but it still does not appear to have met the standard of Nature Communications in terms of originality and broad interest.

1. The authors argued that the toy chaotic electronic circuit system is a real-world system. It is not as it is essentially a circuit realization of a set of differential equations.

2. Stabilizing an arbitrarily long chaotic trajectory through control was published in 1994: PRL 72, 1647 (1994) "Synchronization of chaotic diode resonators by occasional proportional feedback." This diode resonator system has a much fast time scale and is much more "real world" than authors' toy circuit. (The authors mixed it with the works on targeting chaotic trajectories.)

3. A power consumption comparison should be made with that in the above 1994 chaotic diode-resonator paper.

4. In the field of chaos control, scaling relations are extremely important because of the random nature of the dynamics. This has been done in many well known works on chaos control. In my previous report, I provided a relatively detailed prescription of how such a scaling relation can be obtained. This, of course, would require substantially more work than reported in the paper. It seems that the authors did not wish to spend more effort in this work and instead argued that obtaining a scaling relation is not essential. I respectfully disagree: for chaos control scaling relations are of great physical importance.

5. The toy circuit studied is a low-dimensional chaotic system. Demonstrating successful control of a high-dimensional system is challenging but seems essential for consideration by a high-impact journal. The authors stated that this will be their future work. Perhaps a future manuscript can be considered by Nature Communications, but present one does not meet the journal acceptance criteria.

Reviewer 1:

1.1: In their revised version the authors carefully addressed all my questions and significantly changed the manuscript. After reading the reports I am convinced that the paper now meets the requirement of novelty and broad interest and I suggest publication in its present form.

We thank the reviewer for their time and for recommending the manuscript for publication. Reviewer 2:

2.1: The revised manuscript has addressed all the issues that I have raised in the previous round, and I would recommend publication.

We thank the reviewer for their time and for recommending the manuscript for publication. Reviewer 3:

3.1: I appreciate the authors' effort to rebut the referee comments and to revise the paper by addressing some of these comments. The revised paper has been improved to some extent, but it still does not appear to have met the standard of Nature Communications in terms of originality and broad interest.

We thank the reviewer for their time and additional comments. Regarding the standard for originality and broad interest, we note that reviewer #1 explicitly states the manuscript now meets these requirements and that both reviewer #1 and reviewer #2 have recommended the manuscript for publication.

3.2: The authors argued that the toy chaotic electronic circuit system is a real-world system. It is not as it is essentially a circuit realization of a set of differential equations.

In the response to our previous submission, we provided multiple reasons as to why controlling a chaotic electronic circuit serves as a real-world system, such as the presence of chaotic dynamics that increases the sensitivity to the control loop latency as well as the timescale of the

system being shorter than other typical real-world examples. However, we recognize that the reviewer's criteria for a real-world system is different than ours. To this end, we note that the manuscript does not claim to control a real-world system, rather we use the term "difficult control problem". Applying our approach to other real-world examples is part of our future work.

Regarding the statement that the chaotic electronic circuit is a circuit realization of a set of differential equations, we note that this system is a member of the Chua's circuit family, which was the first autonomous circuit discovered to exhibit chaos and has been the subject of hundreds of papers over the past three decades. The long-lasting interest in this circuit lies not in its role as a differential equation solver, but rather because it is a simple system that displays complex and chaotic behavior. Demonstrating the effectiveness of our control algorithm on such a well-studied example of chaos is more impactful than applying it to a system whose differential equations are unknown.

We also note that there are reports of systems that are explicitly designed to solve a set of differential equations using analog circuits. These often involve analog multipliers for generating nonlinear terms and operational amplifiers for performing integration of a voltage. Our double-scroll system does not fall into this class of systems and hence is just as "real" as the circuit mentioned by the referee.

3.3: Stabilizing an arbitrarily long chaotic trajectory through control was published in 1994: PRL 72, 1647 (1994) "Synchronization of chaotic diode resonators by occasional proportional feedback."

We would like to point out that synchronization and control are different problems, and each comes with unique challenges. That being said, the work by Newell *et al.* 1994 mentions that a computer with a digital to analog converter can supply the master signal to synchronize a single resonator rather than synchronizing two physical chaotic resonators. As the reviewer mentions, this should allow the user to control the resonator to an arbitrarily long chaotic trajectory, so long as it is embedded in the attractor. No evidence is given for controlling the system to arbitrary trajectories that are not embedded in the attractor, which we demonstrate with the random waveform trajectory. Furthermore, you can have a large control-loop latency when synchronizing two systems; this will just give rise to lagged synchronization. On the other hand, large control-loop latency will destabilize closed-loop feedback, another point demonstrating that synchronization is a different control problem in comparison to closed-loop feedback control.

In the occasional proportional feedback (OPF) approach published by E.R. Hunt in PRL 67 (1991), controlling the resonator to different orbits required manually tuning parameters such as the feedback gain depending on the location of the system in phase space when the measurements are performed, or perturbations applied. Like the approach by Ott, Grebogi, and Yorke (OGY) in 1990, the OPF approach relies on a linear approximation of the system's

dynamics in the vicinity of the fixed-points or orbits. This approach is not suitable for controlling to a truly arbitrary trajectory where nonlinear effects are unavoidable.

On the other hand, our approach does not require manual tuning of parameters when changing the desired state because the reservoir computer learns the dynamics of the underlying system across the entire phase space.

3.4: This diode resonator system has a much fast time scale and is much more "real world" than authors' toy circuit. (The authors mixed it with the works on targeting chaotic trajectories.)

While the chaotic diode resonator does have a faster timescale than the circuit we study, the rate at which our controller can apply feedback is primarily limited by the latency of the analog to digital converters (ADCs) and digital to analog converters (DACs), which is two orders of magnitude longer than the time it takes to evaluate the control law. Control of systems with much faster timescales should be possible using faster FPGAs with lower latency analog inputs and outputs, but the size, weight, and power of these devices is considerably higher. The important point is that we matched the resources available on the FPGA with the system to be controlled.

3.5: A power consumption comparison should be made with that in the above 1994 chaotic diode-resonator paper.

Due to reasons stated in our response to comment 3.3, we believe this comparison is not necessary. In addition, we note that the editor and other reviewers have decided that obtaining scaling relations (related to power consumption) goes beyond the scope of this work.

3.6: In the field of chaos control, scaling relations are extremely important because of the random nature of the dynamics. This has been done in many well known works on chaos control. In my previous report, I provided a relatively detailed prescription of how such a scaling relation can be obtained. This, of course, would require substantially more work than reported in the paper. It seems that the authors did not wish to spend more effort in this work and instead argued that obtaining a scaling relation is not essential. I respectfully disagree: for chaos control scaling relations are of great physical importance.

We agree that scaling relations are important and thank the referee again for providing a detailed prescription of how it can be obtained. However, in the previous letter, the referee had emphasized that the reason for obtaining such a scaling relation was so that it could be compared to the OGY approach, which we argued was not essential because the OGY approach cannot be used to control the chaotic circuit to all the trajectories we study here. We stand by our original statement and again note that the editors and other reviewers decided that obtaining scaling relations is outside the scope of this work.

However, we can offer some insight on how the power consumption scales when using the FPGA platform specifically. As stated in the manuscript, most of the power consumption comes from quiescent power consumption and the ADCs and DACs. The power consumption increases approximately quadratically with the sampling rate and linearly with the number of ADCs and DACs, which may be necessary for systems with faster timescales and higher dimensionality, respectively.

3.7: The toy circuit studied is a low-dimensional chaotic system. Demonstrating successful control of a high-dimensional system is challenging but seems essential for consideration by a high-impact journal. The authors stated that this will be their future work. Perhaps a future manuscript can be considered by Nature Communications, but present one does not meet the journal acceptance criteria.

As stated in our previous rebuttal, the control algorithm requires only an accurate model of the system, and it has been previously shown that a next-generation reservoir computer can learn accurate models of high-dimensional chaotic systems, such as the extended Lorenz 96 system (see reference 34). Thus, our approach should apply to higher dimensional systems given an FPGA device with more analog inputs and outputs. Additionally, we note that the editor and other reviewers have agreed that obtaining new results for high-dimensional systems is beyond the scope of this work.

Additional changes

We have made small editorial changes throughout the text, which are listed below:

- We have changed the affiliation for Daniel Gauthier to
“²ResCon Technologies, LLC, 1275 Kinnear Rd., Suite 239, Columbus, OH 43212, USA”
- We have replaced the shaded regions in Supplementary Figure 2g-I with hatched lines to improve clarity and to meet compatibility requirements for the “.eps” file type.
- We have removed a duplicate citation (references 10 and 17 in the previous submission are now just reference 10 by Modha *et al.* 2023).
- We have updated multiple citations to conform to the Nature style standards (references 1,29, and 35)
- We have updated the citation to the preprint by Kent *et al.* 2023 to the recently published version (reference 23)
- We have included a citation for the fxpmath Python package (reference 38)
- We have changed the following text from
“To generate the perturbation, we use uniform random noise in Python using `numpy.random.uniform`³⁶ in the range ± 1.5 with zero mean. This data is processed by a 10th-order low-pass Butterworth filter using `scipy.signal.butter`³⁷ with a sample rate 200 kHz and cut-off frequency of 900 Hz.”
to
“To **create** the perturbations, we **generate uniform random noise with a range of ± 1.5 and zero mean using the `random.uniform` package in NumPy.**³⁶ We then filter the noise

using a 10th-order low-pass Butterworth filter with a cut-off frequency of 900 Hz, which is implemented using the `signal.butter` package in SciPy.³⁷

where the references to the documentation of the `random.uniform` [36] and `signal.butter` [37] packages have been replaced with references to the NumPy and SciPy libraries, respectively. This change was made to conform to the NumPy and SciPy citing suggestions available at <https://projects.scipy.org/citing.html>.

- We have changed the following text:
“The ridge parameter for each of the 5 datasets is chosen from the values generated by `numpy.logspace(-10,1,100)`³⁸ to minimize the prediction RMSE on the validation set.”
to
“The ridge parameter for each of the 5 datasets is chosen from an array of 100 logarithmically spaced values from 10^{-10} to 10 to minimize the prediction RMSE on the validation set.”
as we believe this wording will be clearer for a general audience.
- We have added a Data and Code Availability Statements to the manuscript, as seen here:

Data Availability

The data used to create the figures in this manuscript and the Supplementary Material are available in figshare with the identifier “doi: 10.6084/m9.figshare.25534552”.³⁹

Code Availability

The Python and FPGA code with example data are available in figshare with the identifier “doi: 10.6084/m9.figshare.25534621”.⁴⁰

and the associated references 39 and 40 have been added to the manuscript, shown here

39. Kent, R. M., Barbosa, W. A. S. & Gauthier, D. J. Controlling Chaos Using Edge Computing Hardware Data sets. *figshare* <https://doi.org/10.6084/m9.figshare.25534552> (2024).

40. Kent, R. M., Barbosa, W. A. S. & Gauthier, D. J. Controlling Chaos Using Edge Computing Hardware code. *figshare* <https://doi.org/10.6084/m9.figshare.25534621> (2024).

The data and code are now publicly available, which can be accessed via the corresponding URLs in the references above.